# The impact of a randomized controlled trial of a lifestyle intervention on postpartum physical activity among at-risk hispanic women: Estudio PARTO

Sarah Burkart[1], Bess H. Marcus[2], Penelope Pekow[3], Milagros C. Rosal[4], JoAnn E. Manson[5], Barry Braun[6], Lisa Chasan-Taber[3]*

1 Department of Kinesiology, School of Public Health & Health Sciences, University of Massachusetts, Amherst, Massachusetts, United States of America, 2 Department of Behavioral and Social Sciences, Brown University School of Public Health, Providence, Rhode Island, United States of America, 3 Department of Biostatistics and Epidemiology, School of Public Health & Health Sciences, University of Massachusetts Amherst, Amherst, Massachusetts, United States of America, 4 Division of Preventive and Behavioral Medicine, Department of Population & Quantitative Health Sciences, University of Massachusetts Medical School, Worcester, Massachusetts, United States of America, 5 Department of Medicine, Brigham and Women's Hospital, Harvard Medical School, Boston, Massachusetts, United States of America, 6 Department of Health and Exercise Science, College of Health and Human Sciences, Colorado State University, Fort Collins, Colorado, United States of America

* LCT@umass.edu

**Data Availability Statement:** Research resources generated with funds from this grant will be freely

## Abstract

### Aims

To assess the impact of a culturally modified, motivationally targeted, individually-tailored intervention on postpartum physical activity (PA) and PA self-efficacy among Hispanic women.

### Methods

Estudio PARTO was a randomized controlled trial conducted in Western Massachusetts from 2013–17. Hispanic women who screened positive for gestational diabetes mellitus were randomized to a Lifestyle Intervention (LI, n = 100) or to a comparison Health and Wellness (HW, n = 104) group during late pregnancy. Exercise goals in LI were to meet American College of Obstetrician & Gynecologists guidelines for postpartum PA. The Pregnancy Physical Activity Questionnaire (PPAQ) and the Self-Efficacy for Physical Activity Questionnaire were administered at 6 weeks, 6 months, and 1 year postpartum.

### Results

Compared to baseline levels, both groups had significant increases in moderate-to-vigorous PA at 6 months and one year postpartum (i.e., LI: mean change = 30.9 MET-hrs/wk, $p$ = 0.05; HW: 27.6 MET-hrs/wk, $p$ = 0.01), with only LI group experiencing significant increases in vigorous PA (mean change = 1.3 MET-hrs/wk, $p$ = 0.03). Based on an intent-to-treat analysis using mixed effects models, we observed no differences in pattern of change in PA

distributed, as available, to qualified academic investigators for non-commercial research. Our institution (University of Massachusetts) will adhere to the NIH Grants Policy on Sharing of Unique Research Resources including the "Sharing of Biomedical Research Resources: Principles and Guidelines for Recipients of NIH Grants and Contracts" issued in December, 1999; https://grants.nih.gov/grants/intell-property_64FR72090.pdf.

**Funding:** LCT, BHM, PP, MCR, JEM, BB received funding from NIH/NIDDK DK064902. The funders had no role in study design, data collection and analysis, decision to publish, or preparation of the manuscript.

**Competing interests:** The authors have declared that no competing interests exist.

intensity and type over time between intervention groups (all $p > 0.10$). However, there was the suggestion of a greater decrease in sedentary activity in the LI group compared to the HW group ($\beta = -3.56$, $p = 0.09$).

## Conclusions

In this randomized trial among high-risk Hispanic women, both groups benefitted from participation in a postpartum intervention.

## Introduction

Globally, the number of people with diabetes is projected to rise from 382 million in 2013 to 592 million in 2030 [1]. Simultaneously, the age at onset of type 2 diabetes is decreasing which highlights the importance of identifying high-risk groups early in order to implement prevention efforts [2]. Women who develop gestational diabetes mellitus (GDM) or milder glucose intolerance in pregnancy are at high risk for type 2 diabetes [3, 4]; and women who develop GDM in pregnancy have a 7-fold higher risk for developing type 2 diabetes in the future [5].

According to a systematic review, the highest risk period for the development of type 2 diabetes is within the first 5 years after a diagnosis of GDM [3]; with 50% of Hispanic women developing type 2 diabetes during this time period [6]. This is consistent with findings showing a rapid postpartum change in glucose tolerance among women with GDM and with milder degrees of progressing to prediabetes or diabetes as early as one year postpartum [7]. Thus, the pregnancy and postpartum periods represent critical opportunities for interventions designed to prevent subsequent onset of diabetes in susceptible populations.

Currently, the American College of Obstetricians and Gynecologists (ACOG) recommends that postpartum women attain 150 minutes of moderate intensity physical activity (PA) in the absence of any delivery complications [8]. However, postpartum PA levels among women with GDM rarely meet these recommendations [9] and, among Hispanic postpartum women, 51% reported being inactive and only 11% reported engaging in moderate-intensity PA, substantially lower rates as compared to non-Hispanic whites [10]. For women with young families, the lack of time, cost of childcare, physical limitations, and feeling tired are major barriers to participation in PA [11]. Low PA following pregnancy may also be due to lack of PA self-efficacy, which is an individual's confidence in their ability to be physically active [12].

Many lifestyle interventions designed to increase PA among non-pregnant adults at risk of diabetes have had a positive impact on reducing risk of type 2 diabetes in general population samples [13] and among US Hispanic adults as well [14]. However, lifestyle interventions designed to increase PA among postpartum women are sparse and among the few that targeted women with previous GDM [15–19] many occurred after the index pregnancy [17–19]. In contrast, lifestyle interventions that start during pregnancy and continue postpartum may be more effective by taking advantage of the "teachable moment" of pregnancy. In addition, many prior postpartum lifestyle interventions required travel to a gym or clinic and did not utilize a theoretical framework to guide intervention development. Few studies measured the impact of their interventions on PA self-efficacy.

Finally, few postpartum interventions specifically targeted at-risk Hispanic women. Hispanic women have higher rates of overweight and obesity, excessive gestational weight gain, and GDM as compared to non-Hispanic white women [20]. This is critical as Hispanics are the largest and fastest growing ethnic group in the US with the percentage of Hispanic women

of childbearing age projected to increase 74% by 2060, in contrast to an expected 35% decline in non-Hispanic whites over the same time period [21]. In addition, as compared to non-Hispanic white women, Hispanic women with a history of GDM are less aware of diabetes risk factors and prevention strategies (e.g., PA, dietary behaviors, and weight management) [22]. However, despite these findings, socioeconomic, health literacy, and language barriers have limited the access of Hispanic women to interventions that promote healthy lifestyles.

Therefore, our aims were to examine the effect of a lifestyle intervention on postpartum PA levels and PA self-efficacy in at-risk Hispanic women who screened positive for GDM. The intervention was designed to be a low-cost high-reach strategy such that findings could readily be translated into clinical practice in underserved and minority populations. We hypothesized that women randomized to a Lifestyle Intervention (LI) group would report less sedentary time and greater moderate and vigorous intensity PA and sports/ exercise), and would be more likely to meet ACOG PA guidelines [8] compared to those randomized to a comparison Health and Wellness (HW) Intervention. We also hypothesized that women randomized to LI would demonstrate greater PA self-efficacy compared to women randomized to HW.

## Materials and methods

### Study design

Estudio PARTO (Project Aiming to Reduce Type two diabetes) study [23] was a randomized controlled trial conducted from January 2013 to December 2017 at Baystate Medical Center in Western Massachusetts to test the efficacy of a culturally and linguistically modified, individually-tailored lifestyle intervention to reduce risk factors for type 2 diabetes and cardiovascular disease, including low levels of PA and self-efficacy, among postpartum Hispanic women who screened positive for GDM. The study was approved by the University of Massachusetts Amherst and Baystate Health Institutional Review Boards (IRB reference #: BH-12-111).

Eligible women were recruited by bilingual/bicultural health educators at the time of routine screening for GDM (24–28 weeks gestation) and randomly assigned to either a LI or HW Intervention (Fig 1). Randomization was stratified by the results of the diagnostic 100 g oral glucose tolerance test (OGTT) using thresholds defined by the American Diabetes Association (ADA): 1) no glucose values meeting or exceeding the ADA thresholds; or 2) one or more glucose values meeting or exceeding the ADA thresholds) [24].

In both intervention groups, an introductory phase (~29 weeks gestation to the time of birth) was followed by an active phase (6 weeks postpartum– 6 months postpartum) and then a maintenance phase (6 months–one year postpartum). Both the introductory phase and the active phase were kicked-off with an in-person session, after which participants received weekly, biweekly, and monthly telephone booster calls and mailings. Tailoring questionnaires administered at 6 weeks, 6 months, and one year postpartum were used to inform culturally-tailored and motivational individually-tailored feedback. All print-based intervention materials were in Spanish or English depending upon participant preference.

Trained bicultural/bilingual health interviewers, blinded to the intervention group, conducted assessments at baseline, and at 6-weeks, 6-months, and 12-months postpartum.

### Study population

Eligibility was limited to Hispanic women at elevated risk for developing type 2 diabetes, defined by having an abnormal result (i.e., $\geq 135$ mg/dL) on the GDM screen. Hispanic ethnicity was identified via self-report by asking, "Are you Hispanic or of Spanish or Hispanic origin or descent? (response options: yes, no)" in the manner of the U.S. Census.

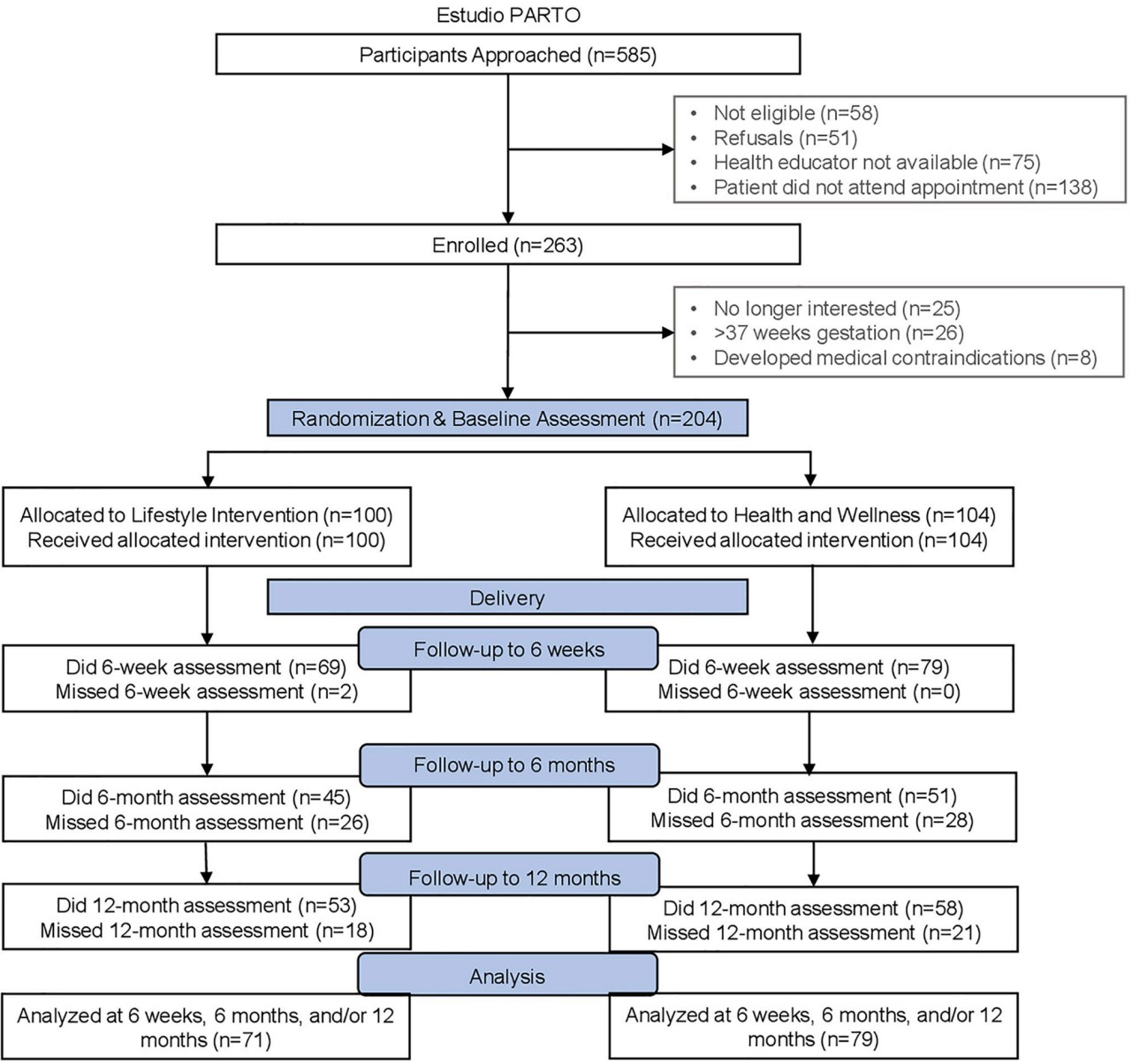

**Fig 1. Flow chart of participant recruitment and retention; Estudio PARTO 2013–2016.**

Women were excluded if they had: 1) a history of type 1 or type 2 diabetes, heart disease, or chronic renal disease, 2) <16 or >45 years of age, 3) contra-indications to engaging in moderate PA and consuming a low fat/high fiber diet, or 4) self-reported inability to read English or Spanish at a sixth grade level.

A total of 585 participants were approached and screened for eligibility (Fig 1). Of these, 263 women met the eligibility criteria and were enrolled in the study. Fifty-nine were subsequently excluded prior to randomization because they were no longer interested (n = 25), >37 weeks gestation (n = 26), or developed other medical contraindications (n = 8). The remaining 204 women were randomized to LI (n = 100) or to HW Intervention (n = 104). Of these women, 54 chose not to participate in the study after delivery, and thus were excluded due to

missing PA data at all three postpartum time points. Therefore, the final sample included 150 women: LI (n = 71) and HW Intervention (n = 79).

## Lifestyle intervention

The LI was designed by utilizing constructs from both Social Cognitive Theory [12] and the Transtheoretical Model [25] and took into account the social, cultural, economic, and environmental resources and challenges faced by Hispanic women [26, 27]. During the introductory phase, the goal of the lifestyle intervention was to shift women along the continuum of precontemplation for PA to contemplation and preparation for the postpartum active intervention phase. The goal of the active phase was to achieve ACOG guidelines for postpartum PA [8]. Participants were asked to choose which type of PA they felt was safe, enjoyable, and easy to incorporate into their daily lives, and were given pedometers and activity logs to help track their weekly PA.

Mailed materials, based upon the tailoring questionnaires, included culturally-tailored stage-matched manuals which emphasized benefits of exercise, how to build social support for PA, and strategies to overcome barriers to PA. Tip sheets included information for women to be physically active with their baby.

Each telephone session involved review of progress towards previous PA goals, problem solving for challenges faced in achieving these goals, and the setting of new PA goals.

## Health and wellness intervention

HW served as the comparison group and received the same number of in-person sessions, telephone calls, and mailings at the same time points as LI. Content was limited to general information available to the public from the ACOG and the American Academy of Pediatrics. These included, for example, mailed booklets on non-exercise and non-dietary general wellness topics (e.g., cigarette smoking) in English or Spanish. These booklets represented high-quality, standard, low-cost self-help material currently available to the public. In this manner, we controlled for contact time while keeping the content of the two interventions distinct [23].

## Outcome assessment

**Postpartum physical activity levels.** The Pregnancy Physical Activity Questionnaire (PPAQ) was used to measure pregnancy and postpartum PA. The PPAQ is a semi-quantitative instrument, previously shown to be valid and reliable in this population, that measures the duration and intensity of time spent in household/caregiving, occupational, transportation, and sports/exercise activities [28]. The number of minutes spent in each reported activity was multiplied by its metabolic equivalent of task (MET) level and summed to arrive at an estimate of average weekly MET-hours/week.

We calculated the average number of MET-hrs/wk spent in each type of PA (i.e., household/caregiving, transportation, occupational, and sports/exercise) and in each intensity-level of PA (i.e., sedentary [<1.5 METs], light [1.5–<3.0 METs], moderate to vigorous [MVPA ≥3 METs], and vigorous intensity PA [≥6 METS]). Meeting ACOG guidelines was categorized as ≥ 7.5 MET hours/week of moderate-vigorous sports and exercise [8].

**Self-efficacy.** Self-efficacy for PA was measured using the 5-item scale Self-Efficacy for Physical Activity Questionnaire [29] that contains five items designed to assess participants' confidence in their ability to exercise despite various situations. Participants were asked to rate their level of confidence on a 5-point scale with higher scores indicating greater self-efficacy. We also calculated the self-efficacy t-score which rescales the raw score into a standardized score with a mean of 50 and a standard deviation (SD) of 10.

The Self-Efficacy for Physical Activity Questionnaire has adequate internal consistency, test–retest reliability, and concurrent validity and has been found to be reliable and valid among a sample of Hispanic women [29].

## Covariate assessment

Demographic and behavioral characteristics were collected at baseline via standardized questionnaires. Sociodemographic characteristics included age, education, marital status, annual household income, living situation, number of adults and children in the household, and generation in the continental US. Acculturation was assessed with the Psychological Acculturation Scale (PAS) [30]. PAS scores between 1 and <3 were categorized as low acculturation, and scores ≥3 were categorized as high acculturation. Sleep was assessed using the Pittsburgh Sleep Quality Index (PSQI) [31] as a continuous measure ranging from 1 to 21 with lower scores indicating better quality sleep. Depressive symptoms were assessed with the Edinburgh Postpartum Depression Scale and categorized as at least probable minor depression using the recommended cutpoint of ≥13 [32].

Clinical characteristics of the pregnancy were abstracted from the medical record and included parity and prepregnancy BMI. Glucose tolerance in pregnancy was categorized as: 1) isolated hyperglycemia: blood glucose >140mg/dL for the one hour oral glucose challenge test followed by a normal OGTT for all four measures; 2) impaired glucose tolerance: blood glucose >140 mg/dL followed by an OGTT with one abnormal result; and 3) GDM: blood glucose >140 mg/dL followed by an OGTT with ≥2 abnormal results.

## Statistical analyses

Two-sample t-tests and chi square tests were used to explore differences in intervention group characteristics at baseline. Within-group differences in mean changes from baseline (late pregnancy) to 6-weeks, 6 months, and one year postpartum, respectively were assessed with Wilcoxon signed rank tests for continuous variables and with McNemar tests for categorical variables.

We used an intent-to-treat analysis to evaluate the impact of the LI on the adoption and maintenance of increased PA and self-efficacy as compared to the HW group. Specifically, mixed models were used to evaluate differences in the change in PA variables from baseline to one year postpartum between the intervention groups. These models accounted for the repeated measures of PA with random subject effects, and incorporated fixed assessment time effects using baseline PA as the reference. We used generalized estimating equations (proc genmod) to evaluate differences in the percent of women meeting ACOG guidelines between intervention groups over follow-up time.

We then conducted several sensitivity analyses. We limited the analysis to women in LI who were compliant with the intervention defined as: 1) meeting the ACOG guidelines and 2) returning ≥1 tailoring questionnaires during the postpartum period. We repeated the analysis limiting the sample to those with prepregnancy overweight or obesity as some prior studies have found that interventions may be more effective in this population [33]. We also conducted a sensitivity analysis where women who did not attend a postpartum assessment visit were categorized as not having met ACOG guidelines at that time point. Finally, we compared those missing all PA data to those with PA data at any time point.

All analyses were considered statistically significant at an alpha <0.05. SAS version 9.3 (SAS Institute Inc.) was used for all statistical analysis.

## Results

Participants had an average age of 27.7 ± 5.7 years, mean prepregnancy BMI of 31.0±7.7 kg/m$^2$, and the majority were in the obese category (Table 1). One-third (32.4%) of the participants were married and the majority reported living with a spouse or partner (67.6%). Approximately one third of the participants had been diagnosed with GDM. There were no statistically significant differences between groups in sociodemographic factors or medical history variables, however the LI group had more mild forms of glucose intolerance compared to the HW group ($p$ = 0.003).

Baseline PA levels did not vary significantly between study groups (Table 2). The majority of women in the sample were not active, with 36.6% of participants in LI and 31.7% of participants in HW meeting ACOG guidelines for exercise in pregnancy.

We then evaluated within-group changes in PA from baseline to 6 weeks, 6 months, and one year postpartum (Table 3). In LI, mean sedentary MET hours/week decreased from baseline to 6 months (-6.7 ± 14.0, $p$ = 0.004) and to one year (-5.3 ± 2.2, $p$ = 0.018), and MVPA increased significantly from baseline to each follow-up time period. For example, at one year postpartum, the mean increase in MVPA was 30.9 ± 107.9 MET hrs/wk as compared to baseline ($p$<0.05). Household activity increased from baseline to 6 weeks (44.7 ± 87.4 MET hrs/wk, $p$<0.0001) while occupational activity decreased over this time frame (-22.2 ± 67.1 MET hrs/wk, $p$ = 0.01). There were no other significant changes in the other PA domains, nor in the percent of participants meeting ACOG PA guidelines.

In contrast, HW reported a significant overall increase in total MET hrs/wk from baseline to 6 weeks postpartum (47.5 ± 139.4 MET hrs/wk, $p$ = 0.01) but not to later follow-up time points (Table 3). Other domain specific changes were similar to LI. For example, mean sedentary MET hours/week decreased from baseline to 6 months (-3.9 ± 13.0, $p$ = 0.01) and to one year (-5.4 ± 14.3, $p$ = 0.002). MVPA increased significantly from baseline to each follow-up time period; for example, at 1 year postpartum, the mean increase in MVPA was 27.6 ± 87.8 MET hrs/wk as compared to baseline ($p$ = 0.01). Household activity increased from baseline to 6 weeks (59.0 ± 94.2 MET hrs/wk, $p$<0.0001) and to 6 months, while occupational activity decreased over this time frame (-20.6 ± 61.5 MET hrs/wk, $p$ = 0.002). Unlike LI, HW reported an increase in transportation activity from baseline to 6 weeks (6.1 ± 34.3 MET hrs/wk, $p$ = 0.03).

In models including interaction terms to evaluate group differences in change in PA intensity and type from baseline to one year postpartum, we observed no differences in pattern of change over time between intervention groups (all $p$>0.10). After removing interaction terms, there was the suggestion of a greater decrease in sedentary activity in LI compared to HW (β = -3.56, $p$ = 0.09) (Table 4). We then evaluated differences in the percent of women meeting ACOG guidelines between intervention groups over time. We found the suggestion that the LI group was less likely to meet guidelines as compared to HW, however this was not statistically significant (OR = 0.61, 95% CI 0.36–1.04).

We then evaluated the impact of the intervention on PA self-efficacy. While LI reported a significant overall increase in self-efficacy from 6 weeks to 6 months postpartum (mean change in t score = 2.57, $p$ = 0.04), HW reported a non-significant decrease (-0.96, $p$ = 0.38). However findings from mixed models yielded no greater improvement over time in self-efficacy in LI vs HW from baseline to one year postpartum ($p$ = 0.31).

We then conducted several sensitivity analyses. Findings were virtually unchanged after limiting the analysis to women who were compliant with LI defined as meeting the ACOG guidelines at any postpartum time point (30.8–42.0% of the LI group), or after limiting the analysis to participants in either group who returned one or more tailoring questionnaires

**Table 1. Distribution of covariates according to intervention arm; Estudio PARTO 2013–2017.**

| Covariate | Lifestyle Intervention (N = 100) | | Health & Wellness Intervention (N = 104) | |
|---|---|---|---|---|
| | N or Mean | % or SD | N or Mean | % or SD |
| Age | | | | |
| 16–19 | 7 | 7.0% | 8 | 7.7% |
| 20–24 | 18 | 18.0% | 29 | 27.9% |
| 25–29 | 39 | 39.0% | 35 | 33.7% |
| 30–34 | 25 | 25.0% | 14 | 13.5% |
| 35–45 | 11 | 11.0% | 18 | 17.3% |
| Education | | | | |
| Less than high school | 25 | 25.0% | 27 | 26.0% |
| High school graduate or GED | 28 | 28.0% | 32 | 30.8% |
| Post high school | 47 | 47.0% | 45 | 43.3% |
| Marital Status | | | | |
| Single/Separated/Divorced/Widowed | 72 | 72.0% | 77 | 74.0% |
| Married | 28 | 28.0% | 27 | 26.0% |
| Live with Partner or Spouse | | | | |
| No | 32 | 32.0% | 27 | 26.2% |
| Yes | 68 | 68.0% | 76 | 73.8% |
| Annual Household Income | | | | |
| < = $15,000 | 16 | 16.0% | 30 | 28.9% |
| >$15,000-$30,000 | 15 | 15.0% | 15 | 14.4% |
| >$30,000 | 20 | 20.0% | 17 | 16.4% |
| missing (don't know/refused) | 49 | 49.0% | 42 | 40.4% |
| Adults in Household | | | | |
| 0 | 0 | 0.0% | 1 | 1.0% |
| 1 | 16 | 16.0% | 8 | 7.7% |
| 2 | 56 | 56.0% | 64 | 61.5% |
| 3 or more | 28 | 28.0% | 31 | 29.8% |
| Children in Household | | | | |
| 0 | 20 | 20.0% | 22 | 21.2% |
| 1 | 44 | 44.0% | 38 | 36.5% |
| 2 | 22 | 22.0% | 23 | 22.1% |
| 3 or more | 14 | 14.0% | 21 | 20.2% |
| Generation | | | | |
| Born outside continental US | 47 | 47.5% | 43 | 41.4% |
| Parent born outside continental US | 32 | 32.3% | 43 | 41.4% |
| Grandparent born outside continental US | 14 | 14.1% | 10 | 9.6% |
| All grandparents born outside continental US | 6 | 6.1% | 8 | 7.7% |
| Acculturation Status | | | | |
| Low acculturation (1 to <3) | 78 | 78.0% | 76 | 73.1% |
| High acculturation (> = 3) | 22 | 22.0% | 28 | 26.9% |
| Total Sleep Score (mean, SD) | 6.9 | 4.3 | 6.8 | 4.2 |
| At Least Probable Minor Depression | | | | |
| No | 62 | 93.9% | 59 | 93.7% |
| Yes | 4 | 6.1% | 4 | 6.4% |
| Prepregnancy smoking | | | | |
| No | 74 | 74.0% | 79 | 76.0% |
| up to 10 per day | 25 | 25.0% | 22 | 21.2% |

(*Continued*)

**Table 1.** (Continued)

| Covariate | Lifestyle Intervention (N = 100) | | Health & Wellness Intervention (N = 104) | |
|---|---|---|---|---|
| | N or Mean | % or SD | N or Mean | % or SD |
| Over 10 per day | 1 | 1.0% | 3 | 2.9% |
| Smoked during pregnancy | | | | |
| No | 93 | 93.0% | 99 | 95.2% |
| Up to 10 per day | 7 | 7.0% | 5 | 4.8% |
| Over 10 per day | 0 | 0.0% | 0 | 0.0% |
| Parity | | | | |
| 0 | 31 | 31.0% | 23 | 22.1% |
| 2 to 3 | 54 | 54.0% | 62 | 59.6% |
| 3 or more | 15 | 15.0% | 19 | 18.3% |
| BMI | | | | |
| <18.5 | 2 | 2.0% | 0 | 0.0% |
| 18.5 - <25 | 23 | 23.0% | 23 | 22.1% |
| 25 - <30 | 26 | 26.0% | 37 | 35.6% |
| > = 30 | 49 | 49.0% | 44 | 42.3% |
| Glucose Tolerance | | | | |
| Isolated hyperglycemia | 38 | 38.0% | 53 | 51.0% |
| Impaired glucose tolerance | 30 | 30.0% | 11 | 10.6% |
| Gestational diabetes mellitus | 32 | 32.0% | 40 | 38.5% |

during the postpartum period (42.9% in LI and 57.1% in HW). Findings were similarly unchanged after categorizing women who did not attend postpartum assessment visits as not having reached ACOG guidelines. When we repeated the main analysis among women with prepregnancy overweight or obesity (78.9% in LI, and 76.0% in HW), findings were also virtually unchanged.

**Table 2.** Baseline physical activity according to intervention group (n = 150); Estudio PARTO 2013–2017.

| | Lifestyle Intervention (N = 71) | | Health & Wellness Intervention (N = 79) | | P value |
|---|---|---|---|---|---|
| | N | % | N | % | |
| Met ACOG guidelines* | | | | | |
| No | 45 | 63.4% | 54 | 68.4% | 0.52 |
| Yes | 26 | 36.6% | 25 | 31.7% | |
| Total MET hrs/wk (mean, SD) | 220.9 | 106.9 | 219.2 | 90.3 | 0.92 |
| Physical activity intensity (MET hrs/wk) | | | | | |
| Sedentary | 16.3 | 14.2 | 13.8 | 11.7 | 0.26 |
| Light intensity | 132.3 | 61.4 | 128.0 | 53.8 | 0.65 |
| Moderate intensity | 72.5 | 72.6 | 74.0 | 65.6 | 0.89 |
| Vigorous intensity | 0.5 | 1.6 | 0.6 | 1.9 | 0.81 |
| Moderate + vigorous intensity | 73.0 | 72.6 | 74.6 | 65.6 | 0.89 |
| Physical activity type (MET hrs/wk) | | | | | |
| Household | 105.7 | 71.8 | 115.9 | 66.7 | 0.37 |
| Occupational | 56.6 | 59.6 | 49.5 | 65.5 | 0.49 |
| Sports/exercise | 6.0 | 7.1 | 6.9 | 8.3 | 0.47 |
| Transportation | 28.2 | 25.4 | 21.2 | 22.4 | 0.07 |

ACOG = American College of Obstetricians and Gynecologists; MET = metabolic equivalent.

* ≥ 7.5 MET hours/week of moderate-vigorous intensity sports and exercise.

**Table 3. Change in physical activity over follow-up according to intervention group; Estudio PARTO 2013–2017.**

| | Change from Baseline at 6 wks | | | | Change from Baseline at 6 mos | | | | Change from Baseline at 12 mos | | | |
|---|---|---|---|---|---|---|---|---|---|---|---|---|
| | Mean | SD | p-value* | 95% CI | Mean | SD | p-value* | 95% CI | Mean | SD | p-value* | 95% CI |
| **Lifestyle Intervention** | | | | | | | | | | | | |
| Total MET hrs/wk | 25.9 | 118.7 | 0.06 | -4.03, 55.78 | 10.4 | 157.2 | 0.56 | -39.22, 59.99 | 4.9 | 155.6 | 0.89 | -30.83, 50.56 |
| Physical activity intensity (MET hrs/wk) | | | | | | | | | | | | |
| Sedentary | -1.8 | 16.5 | 0.58 | -5.91, 2.23 | -9.2 | 15.4 | 0.0001 | -14.03, -4.31 | -8.2 | 13.6 | <0.0001 | -12.02, -4.30 |
| Light intensity | -11.1 | 80.1 | 0.25 | -30.34, 8.13 | -5.4 | 75.9 | 0.55 | -28.18, 17.43 | -16.3 | 75.3 | 0.14 | -37.24, 4.71 |
| Moderate + vigorous intensity | 29.1 | 75.0 | 0.0003 | 10.64, 47.50 | 24.5 | 107.3 | 0.03 | -7.71, 56.76 | 30.9 | 107.9 | 0.05 | 0.50, 61.21 |
| Vigorous intensity | 1.85 | 5.97 | 0.006 | 0.39, 3.30 | 1.71 | 5.28 | 0.03 | 0.12, 3.29 | 1.28 | 4.34 | 0.03 | 0.06, 2.50 |
| Physical activity type (MET hrs/wk) | | | | | | | | | | | | |
| Household | 44.7 | 87.4 | <0.0001 | 23.72, 65.71 | 11.8 | 105.9 | 0.28 | -20.01, 43.63 | -0.3 | 101.8 | 0.78 | -28.61, 28.06 |
| Occupational | -22.2 | 67.1 | 0.01 | -38.54, -5.79 | 9.8 | 69.0 | 0.35 | -10.91, 30.56 | 22.3 | 83.8 | 0.14 | -0.74, 45.43 |
| Sports/exercise | 1.1 | 12.0 | 0.82 | -1.84, 3.95 | 2.2 | 10.3 | 0.20 | -0.88, 5.33 | 0.8 | 10.0 | 0.60 | -2.05, 3.60 |
| Transportation | -3.4 | 27.3 | 0.25 | -10.09, 3.24 | -1.6 | 31.8 | 0.63 | -11.25, 8.07 | -0.2 | 31.8 | 0.98 | -9.18, 8.86 |
| Met ACOG guidelines (%) | | | | | | | | | | | | |
| switched from 'no' to 'yes' | 11.8% | | 0.1 | | 22.2% | | 0.47 | | 13.7% | | 0.35 | |
| switched from 'yes' to 'no' | 23.5% | | | | 15.6% | | | | 21.6% | | | |
| **Health & Wellness Intervention** | | | | | | | | | | | | |
| Total MET hrs/wk | 47.5 | 139.4 | 0.01 | 15.44, 79.59 | 27.3 | 141.4 | 0.27 | -14.66, 69.31 | 5.4 | 124.0 | 0.96 | -27.82, 38.61 |
| Physical activity intensity (MET hrs/wk) | | | | | | | | | | | | |
| Sedentary | 2.1 | 16.1 | 0.38 | -1.60, 5.75 | -3.9 | 13.0 | 0.01 | -7.65, -0.17 | -5.4 | 14.3 | 0.002 | -9.26, -1.59 |
| Light intensity | -1.8 | 71.3 | 0.73 | -17.77, 14.18 | -12.0 | 72.0 | 0.26 | -32.45, 8.47 | -15.1 | 74.1 | 0.07 | -34.61, 4.36 |
| Moderate + vigorous intensity | 47.4 | 88.9 | <0.0001 | 27.35, 67.42 | 49.1 | 90.9 | 0.002 | 22.97, 75.18 | 27.6 | 87.8 | 0.01 | 4.46, 50.64 |
| Vigorous intensity | 3.61 | 8.56 | 0.003 | 1.68, 5.54 | 2.08 | 6.91 | 0.08 | 0.10, 4.06 | 0.53 | 4.47 | 0.51 | -0.64, 1.71 |
| Physical activity type (MET hrs/wk) | | | | | | | | | | | | |
| Household | 59.0 | 94.2 | <0.0001 | 37.91, 80.09 | 36.0 | 89.9 | 0.01 | 10.45, 61.54 | 2.9 | 84.3 | 0.80 | -19.29, 25.04 |
| Occupational | -20.6 | 61.5 | 0.002 | -34.46, -6.74 | -0.5 | 61.0 | 0.88 | -17.82, 16.85 | 14.2 | 70.7 | 0.10 | -4.42, 32.73 |
| Sports/exercise | 2.4 | 15.0 | 0.61 | -0.98, 5.78 | 2.7 | 14.4 | 0.38 | -1.44, 6.73 | 1.3 | 11.0 | 0.43 | -1.58, 4.20 |
| Transportation | 6.1 | 34.3 | 0.03 | -1.59, 13.78 | 2.7 | 26.1 | 0.11 | -5.09, 10.42 | -3.8 | 26.1 | 0.72 | -10.62, 3.11 |
| Met ACOG guidelines (%) | | | | | | | | | | | | |
| switched from 'no' to 'yes' | 16.7% | | 0.85 | | 20.0% | | 0.64 | | 25.9% | | 0.43 | |
| switched from 'yes' to 'no' | 18.0% | | | | 16.0% | | | | 19.0% | | | |

*p-value from a Wilcoxon signed rank test for continuous variables, McNemar test for categorical variables.

Finally, women missing data did not differ from those in the final sample according to the majority of sociodemographic, behavioral, clinical, and PA variables. However those missing data were less likely to be low income, born outside the continental US, and had lower mean levels of occupational activity than those not missing data ($p<0.05$).

**Table 4. Main effects model: Differences in change in physical activity from baseline over follow-up between intervention groups; Estudio PARTO 2013–2017.**

| | Treatment Effect* (Lifestyle vs. Health & Wellness) | | | |
|---|---|---|---|---|
| | **Difference in LS Means** | **95% CI** | **SE** | **P Value** |
| Total MET hrs/wk | -18.85 | -56.91, 19.22 | 19.26 | 0.33 |
| Physical activity type (MET hrs/wk) | | | | |
| Household | -14.64 | -40.43, 11.14 | 13.05 | 0.26 |
| Occupational | 3.44 | -14.54, 21.14 | 9.10 | 0.71 |
| Sports/exercise | -1.84 | -5.13, 1.45 | 1.67 | 0.27 |
| Transportation | -3.97 | -12.47, 4.53 | 4.30 | 0.36 |
| Physical activity intensity (MET hrs/wk) | | | | |
| Sedentary | -3.56 | -7.72, 0.59 | 2.10 | 0.09 |
| Light intensity | -5.60 | -26.59, 15.38 | 10.62 | 0.60 |
| Moderate intensity | -16.17 | -40.99, 8.65 | 12.56 | 0.20 |
| Vigorous intensity | -0.23 | -1.64, 1.17 | 0.71 | 0.74 |
| Moderate + vigorous intensity | -17.66 | -42.76, 7.43 | 12.70 | 0.17 |

* Using mixed models.

## Discussion

In this randomized trial of a culturally modified, motivationally targeted, individually-tailored intervention targeting Hispanic women with glucose abnormalities in pregnancy, we did not observe that LI led to a greater improvement in postpartum PA and self-efficacy over HW. However, we did observe the suggestion of a greater decrease in sedentary activity in LI compared to HW. In addition, compared to baseline levels, both groups experienced significant increases in MVPA at 6 months and one year postpartum, with LI only experiencing significant increases in vigorous PA at these time points.

Our findings are comparable to the majority of prior trials among women with GDM that commenced soon after pregnancy and tended to observe no impact of on postpartum PA [17, 19] with only one observing a small increase (e.g., 11 min/day) in leisure time activity [18]. Among trials that began during pregnancy, a Diabetes Prevention Program-derived lifestyle intervention among 2280 participants (22.2% Hispanic) found greater improvements in vigorous intensity PA as measured by the PPAQ in the intervention group compared to the control (mean difference = 15.4 minutes/week, 95% CI = 4.9, 25.8) but no impact on walking, moderate intensity PA, or total PA [16]. In the only study targeting Hispanic women, Hawkins et al. [34] found a slightly higher increase in vigorous PA as measured by the PPAQ at six weeks postpartum in LI compared to the standard care group (0.5 ± 0.5 MET hours/week vs. -0.9 ± 0.5 MET hours/week, *p* = 0.046). Similarly, we found that only women in LI reported a significant increase in vigorous activity from baseline, however this change was not significantly different from HW. These studies did not evaluate the impact of their interventions on sedentary PA.

The significant increases in MVPA reported by both groups were largely due to increases in household/caregiving and occupational activity over the postpartum year, as opposed to sports/exercise, and were consistent with the increased demands of caring for a newborn and returning to work. Specifically, MVPA was made up of 17 types of activity more than half of which were moderate household/caregiving activities, occupational activities or transportation activities.

Few prior studies have evaluated the impact of their intervention on PA self-efficacy [35, 36]. In their clinic-based trial among Taiwanese pregnant women, Huang et al. [36] found significantly greater improvements in postpartum self-efficacy scores in the intervention group

while the Norwegian Fit for Delivery study found no significant differences between groups [35] In the current study, LI reported a significant increase in self-efficacy while HW reported a non-significant decrease however the difference in the change between groups was not significant.

Our findings and those of others indicating that a LI yielded none or only modest improvements in PA among postpartum women with glucose intolerance in pregnancy could be due to several reasons. During pregnancy, factors such as health concerns for the safety of the fetus, physical limitations, and lack resources contribute to poor adherence to PA recommendations [35]. After delivery, the pressures of caring for a new baby and work related obstacles have been reported to be major barriers to an active lifestyle over the postpartum year [37]. Findings that these barriers remain consistent from delivery to 12 months postpartum suggest that they are difficult to overcome [35].

Differences in findings between our study and prior studies could be due to differences in the dose of the lifestyle intervention, the time since the index pregnancy, and in the race/ethnicity of the study samples. Most importantly, differences may be due to our use of a contact-time comparison group that received the same amount and frequency of materials/sessions, enabling us to control for contact time while keeping the content of the interventions distinct. In this manner, we could isolate the impact of LI on PA and rule out the possibility that the social support provided by study staff contributed to differences between groups. Among Hispanic women in particular, social support has been identified as a major facilitator of behavior change [38]. In contrast, previous trials utilized usual care control groups and therefore cannot rule out the role of social support in observed improvements in the lifestyle arm [15, 16, 34, 36].

In addition, unlike some prior studies, we attempted to use a low-cost high reach strategy that involved fewer face-to-face visits and no supervised exercise sessions. While this approach addressed child care and transportation barriers, the intervention dosage may not have been high enough to impact PA one year postpartum. On the other hand, prior studies have reported low compliance rates with supervised exercise sessions; for example, only 16% of participants the in the FitFor2 trial followed at least half of the training sessions [39]. We found that 43–57% of women were compliant in returning tailoring questionnaires, and 31–42% of women in LI met ACOG recommendations. Low levels of adherence may be due to the fact that women with glucose intolerance in pregnancy do not perceive themselves as being at high risk for diabetes after delivery [40]. Given that more than half of our participants (63%) had more mild forms of glucose abnormalities, as opposed to frank GDM, they may have been even less likely to perceive themselves at risk.

Our study faced several limitations. Self-reported assessment of PA may be subject to social desirability bias; however, PA was collected using a validated questionnaire, which minimized potential bias. Participants in this study were not blinded to their intervention group and therefore the LI group may have been more likely to overestimate their exercise. However, total PA included household/caregiving and occupational PA which would be less susceptible to social desirability bias. Approximately one-quarter of women did not participate in the study after delivery and therefore were missing PA data. This group was less likely to be low income and born outside the continental US. To the extent that these factors were associated with postpartum PA, they may have confounded our results. In addition, missing data led to a lower sample size and therefore reduced power to detect differences in change between groups.

## Conclusions

In conclusion, in this trial of a postpartum intervention in Hispanic women, we did not observe that a lifestyle intervention led to a greater improvement in postpartum PA or self-

efficacy. However, women in both intervention groups appeared to benefit from participation in a postpartum intervention. With the growing rates of diabetes and obesity in U.S. women, efforts to improve the effectiveness of lifestyle interventions for the prevention of diabetes in high-risk women becomes critical.

Such interventions should focus on self-selected activities that are sustainable over time and use low-cost strategies such as mail and telephone based interactions to reduce participant burden (e.g., transportation and child care needs). In addition, lifestyle interventions that start during pregnancy would take advantage of the "teachable moment" of pregnancy by addressing the observation that women with glucose intolerance in pregnancy do not perceive themselves as being at high risk for diabetes. Use of such high-reach, low-cost design strategies would facilitate translation of such interventions into clinical practice in underserved and minority populations [41].

## Supporting information

**S1 File. CONSORT 2010 checklist.**
(DOC)

**S2 File. Estudio PARTO study protocol.**
(DOCX)

## Acknowledgments

The authors thank Eva Goldwater and Scott Chasan-Taber for their statistical programming advice.

## Author Contributions

**Conceptualization:** Bess H. Marcus, Penelope Pekow, Milagros C. Rosal, JoAnn E. Manson, Barry Braun, Lisa Chasan-Taber.

**Investigation:** Sarah Burkart, Bess H. Marcus, Penelope Pekow, Milagros C. Rosal, JoAnn E. Manson, Barry Braun, Lisa Chasan-Taber.

**Methodology:** Sarah Burkart, Bess H. Marcus, Penelope Pekow, Milagros C. Rosal, JoAnn E. Manson, Barry Braun, Lisa Chasan-Taber.

**Writing – original draft:** Sarah Burkart, Lisa Chasan-Taber.

**Writing – review & editing:** Sarah Burkart, Bess H. Marcus, Penelope Pekow, Milagros C. Rosal, JoAnn E. Manson, Barry Braun, Lisa Chasan-Taber.

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
