## [Decision Letter · Decision Letter 0]

10 Jun 2020

PONE-D-20-06832

The Impact of a Randomized Controlled Trial of a Lifestyle Intervention on Postpartum Physical Activity among At-Risk Hispanic Women: Estudio PARTO

PLOS ONE

Dear Dr. Chasan-Taber,

Thank you for submitting your manuscript to PLOS ONE. After careful consideration, we feel that it has merit but does not fully meet PLOS ONE’s publication criteria as it currently stands. Therefore, we invite you to submit a revised version of the manuscript that addresses the points raised during the review process.

Specifically, the reviewers suggets some changes in the statistical methods section to clarify some steps taking in analysing the data.

We look forward to receiving your revised manuscript.

Kind regards,

Noël C. Barengo, MD, PhD, MPH

Academic Editor

PLOS ONE

Journal Requirements:

https://www.sciencedirect.com/science/article/pii/S1521693414001618?via%3Dihub

https://bmcpregnancychildbirth.biomedcentral.com/articles/10.1186/1471-2393-14-100

https://bmchealthservres.biomedcentral.com/articles/10.1186/s12913-019-4207-x

In your revision ensure you cite all your sources (including your own works), and quote or rephrase any duplicated text outside the methods section. Further consideration is dependent on these concerns being addressed.

Additional Editor Comments (if provided):

Reviewers' comments:

Reviewer's Responses to Questions

**Comments to the Author**

1. Is the manuscript technically sound, and do the data support the conclusions?

Reviewer #1: Yes

Reviewer #2: Yes

2. Has the statistical analysis been performed appropriately and rigorously? 

Reviewer #1: Yes

Reviewer #2: Yes

3. Have the authors made all data underlying the findings in their manuscript fully available?

Reviewer #1: Yes

Reviewer #2: Yes

4. Is the manuscript presented in an intelligible fashion and written in standard English?

Reviewer #1: Yes

Reviewer #2: Yes

5. Review Comments to the Author

Reviewer #1: Important note: This review pertains only to ‘statistical aspects’ of the study and so ‘clinical aspects’ [like medical importance, relevance of the study, ‘clinical significance and implication(s)’ of the whole study, etc.] are to be evaluated [should be assessed] separately/independently.

This study is excellently planned & executed/conducted and manuscript is nicely drafted (no doubt about that). However, there are few suggestions [may please be considered as they may need to be clarified/told clearly at appropriate places in the text]:

1. In ‘Materials and methods - Study Design’ section it is said that “Randomization was stratified by the results of the diagnostic 100 g oral glucose tolerance test (OGTT)” but ‘what exactly is done’ is neither clear in further text nor from Figure 1 [Flow chart of participant recruitment and retention; Estudio PARTO 2013-2016]. It appears to be simple random allocation (may be by using permuted randomized blocks to ensure nearly same size in groups).

2. This study has taken data (it is a secondary data analysis then – please clarify) from some other trial {Estudio PARTO was a randomized controlled trial conducted in Western Massachusetts from 2013-17} for which the data were collected till 2017 then (my question is) ‘why this delay?’

3. Statistical comparison of baseline characteristics [last ‘p-value’ column in Table 1 (Distribution of Covariates according to Intervention Group; Estudio PARTO 2013-2016)] is not desirable at all [because even if P-value turns out to be significant (while comparing baseline characteristics despite random allocation), it is, by definition, a false positive] as you are supposed to be testing ‘randomization’ then, which in any single trial may not balance all baseline characteristics as ‘randomization’ is a sort of ‘insurance’ and not a guarantee scheme.

(Reference: Harrington D, D’Agostino RB, Gatsonis C, Hogan JW, Hunter DJ, Normand ST, et al. New guidelines for statistical reporting in the journal. N Engl J Med 2019;381:285-6)

4. It is known that ANCOVA has more ‘power’ than ‘change scores’ analyses [Reference: ‘The use of percentage change from baseline as an outcome in a controlled trial is statistically inefficient – a simulation study’ by Andrew Vickers in BMC Medical Research Methodology 2001, 1:6] displayed in table-3, however, it was necessary to use non-parametric tests [like Wilcoxon signed rank test for continuous variables, & McNemar test for categorical variables] on data yielded by tools [like Pregnancy Physical Activity Questionnaire (PPAQ) and the Self-Efficacy for Physical Activity Questionnaire]. Moreover, ANCOVA involves few assumptions which are many times not fulfilled (& are unlikely be fulfilled in this case).

5. Agreed that final analyses included Lifestyle Intervention (N=71) & Health & Wellness Intervention (N=79) but baseline data in table-1 should have all who were randomized to start with [were randomized to a Lifestyle Intervention (LI, n=100) or to a comparison Health and Wellness (HW, n=104) group]. Is not that so? ‘Covariate assessment’ section states that “Demographic and behavioral characteristics were collected at baseline via standardized questionnaires”. ‘Statistical analyses’ section states that “We used an intent-to-treat analysis to evaluate the impact of the LI on the adoption and maintenance of increased PA and self-efficacy as compared to the HW group”.

Otherwise everything else [including Using Mixed Models] is faultless. Except above minor ‘revision’ points, I recommend the publication of this article in our journal.

Reviewer #2: Summary:

This study assess the change in physical activity (PA) of a RCT using an individually-tailored approach for postpartum Hispanic women. The study design and statistics are exceptional. This study is an important addition to the literature because it focuses on Hispanic, postpartum women at high risk for chronic disease and assess the use of a targeted intervention, which is more culturally appropriate and has the potential to advance the field of PA. I applaud these authors for this work and hope to review more studies like this in the future.

Minor Revisions:

Thank you for including your protocol in the additional information. Please include in the text more information on the specific intervention components that differed between the LI and HW groups. You state that your attention control group received the same number of sessions, calls, and mailers – this emphasized the need to know as much information about the difference in interventions as possible.

State future directions for this work, including potential public health interventions that should be considered by practitioners.

6. PLOS authors have the option to publish the peer review history of their article (what does this mean?). If published, this will include your full peer review and any attached files.

Reviewer #1: Yes: Dr. Sanjeev Sarmukaddam

Reviewer #2: No

---

## [Author Response · Author response to Decision Letter 0]

24 Jun 2020

Manuscript PONE-D-20-06832

Editor’s Comments 

1. Specifically, the reviewers suggests some changes in the statistical methods section to clarify some steps taking in analysing the data.

We have revised the statistical methods section to clarify steps taken in analyzing the data as described in the response to reviewer comments below.

We confirm that our manuscript meets PLOS ONE's style requirements including those for file naming. 

https://www.sciencedirect.com/science/article/pii/S1521693414001618?via%3Dihub

https://bmcpregnancychildbirth.biomedcentral.com/articles/10.1186/1471-2393-14-100

https://bmchealthservres.biomedcentral.com/articles/10.1186/s12913-019-4207-x

In your revision ensure you cite all your sources (including your own works), and quote or rephrase any duplicated text outside the methods section. Further consideration is dependent on these concerns being addressed.

We have revised the manuscript to rephrase any duplicated text outside the methods section and now cite our sources, including our own work.

We now include captions for our supporting information files at the end of the manuscript.

Review Comments to the Author 

Reviewer #1: Important note: This review pertains only to ‘statistical aspects’ of the study and so ‘clinical aspects’ [like medical importance, relevance of the study, ‘clinical significance and implication(s)’ of the whole study, etc.] are to be evaluated [should be assessed] separately/independently. This study is excellently planned & executed/conducted and manuscript is nicely drafted (no doubt about that). However, there are few suggestions [may please be considered as they may need to be clarified/told clearly at appropriate places in the text]

We thank the reviewer for this comment and address their suggestions below.

1. In ‘Materials and methods - Study Design’ section it is said that “Randomization was stratified by the results of the diagnostic 100 g oral glucose tolerance test (OGTT)” but ‘what exactly is done’ is neither clear in further text nor from Figure 1 [Flow chart of participant recruitment and retention; Estudio PARTO 2013-2016]. It appears to be simple random allocation (may be by using permuted randomized blocks to ensure nearly same size in groups). 

We have revised the Study Design section (page 6, paragraph 1) to now clarify that randomization was stratified by the results of the diagnostic 100 g oral glucose tolerance test (OGTT) using thresholds defined by the American Diabetes Association (ADA): 1) no glucose values meeting or exceeding the ADA thresholds; or 2) one or more glucose values meeting or exceeding the ADA thresholds) [24]. 

2. This study has taken data (it is a secondary data analysis then – please clarify) from some other trial {Estudio PARTO was a randomized controlled trial conducted in Western Massachusetts from 2013-17} for which the data were collected till 2017 then (my question is) ‘why this delay?’

We apologize for our lack of clarity. We have revised the Study Design section (page 5, paragraph 2) to now clarify that this is a primary data analysis. In other words, one of the primary aims of Estudio PARTO was to evaluate the impact of the intervention on PA and self-efficacy. The reviewer is correct that data was collected on active participants until the end of 2017. We then received a no-cost extension from NIH to allow us additional time to conduct data analysis and interpretation. This, combined with the availability of the coauthors, led to our finalizing the manuscript at the end of 2019 and submitting it in early 2020. 

3. Statistical comparison of baseline characteristics [last ‘p-value’ column in Table 1 (Distribution of Covariates according to Intervention Group; Estudio PARTO 2013-2016)] is not desirable at all [because even if P-value turns out to be significant (while comparing baseline characteristics despite random allocation), it is, by definition, a false positive] as you are supposed to be testing ‘randomization’ then, which in any single trial may not balance all baseline characteristics as ‘randomization’ is a sort of ‘insurance’ and not a guarantee scheme.

(Reference: Harrington D, D’Agostino RB, Gatsonis C, Hogan JW, Hunter DJ, Normand ST, et al. New guidelines for statistical reporting in the journal. N Engl J Med 2019;381:285-6) 

The reviewer makes an excellent point and we have revised Table 1 to no longer include the p-value column. 

4. It is known that ANCOVA has more ‘power’ than ‘change scores’ analyses [Reference: ‘The use of percentage change from baseline as an outcome in a controlled trial is statistically inefficient – a simulation study’ by Andrew Vickers in BMC Medical Research Methodology 2001, 1:6] displayed in table-3, however, it was necessary to use non-parametric tests [like Wilcoxon signed rank test for continuous variables, & McNemar test for categorical variables] on data yielded by tools [like Pregnancy Physical Activity Questionnaire (PPAQ) and the Self-Efficacy for Physical Activity Questionnaire]. Moreover, ANCOVA involves few assumptions which are many times not fulfilled (& are unlikely be fulfilled in this case).

We agree with the reviewer regarding their approval of our change scores analysis and our use of non-parametric tests. 

5. Agreed that final analyses included Lifestyle Intervention (N=71) & Health & Wellness Intervention (N=79) but baseline data in table-1 should have all who were randomized to start with [were randomized to a Lifestyle Intervention (LI, n=100) or to a comparison Health and Wellness (HW, n=104) group]. Is not that so? ‘Covariate assessment’ section states that “Demographic and behavioral characteristics were collected at baseline via standardized questionnaires”. ‘Statistical analyses’ section states that “We used an intent-to-treat analysis to evaluate the impact of the LI on the adoption and maintenance of increased PA and self-efficacy as compared to the HW group”.

We have revised Table 1 to now include all the participants who were randomized to start with (LI, n=100) and (HW, n=104) as per the reviewer’s suggestion.

6. Otherwise everything else [including Using Mixed Models] is faultless. Except above minor ‘revision’ points, I recommend the publication of this article in our journal.

We thank the reviewer for this comment.

Reviewer #2: Summary:

This study assess the change in physical activity (PA) of a RCT using an individually-tailored approach for postpartum Hispanic women. The study design and statistics are exceptional. This study is an important addition to the literature because it focuses on Hispanic, postpartum women at high risk for chronic disease and assess the use of a targeted intervention, which is more culturally appropriate and has the potential to advance the field of PA. I applaud these authors for this work and hope to review more studies like this in the future.

We thank the reviewer for this comment.

Minor Revisions:

1. Thank you for including your protocol in the additional information. Please include in the text more information on the specific intervention components that differed between the LI and HW groups. You state that your attention control group received the same number of sessions, calls, and mailers – this emphasized the need to know as much information about the difference in interventions as possible.

We have revised the Methods section to now include more detail about Health and Wellness intervention received by the attention control group (page 8, paragraph 3).

2. State future directions for this work, including potential public health interventions that should be considered by practitioners.

We have revised the Discussion section to now describe future directions for this work including potential future interventions (page 23, paragraph 2).

---

## [Decision Letter · Decision Letter 1]

8 Jul 2020

The Impact of a Randomized Controlled Trial of a Lifestyle Intervention on Postpartum Physical Activity among At-Risk Hispanic Women: Estudio PARTO

PONE-D-20-06832R1

Dear Dr. Chasan-Taber,

We’re pleased to inform you that your manuscript has been judged scientifically suitable for publication and will be formally accepted for publication once it meets all outstanding technical requirements.

Kind regards,

Noël C. Barengo, MD, PhD, MPH

Academic Editor

PLOS ONE

Additional Editor Comments (optional):

Reviewers' comments:

Reviewer's Responses to Questions

**Comments to the Author**

1. If the authors have adequately addressed your comments raised in a previous round of review and you feel that this manuscript is now acceptable for publication, you may indicate that here to bypass the “Comments to the Author” section, enter your conflict of interest statement in the “Confidential to Editor” section, and submit your "Accept" recommendation.

Reviewer #1: All comments have been addressed

2. Is the manuscript technically sound, and do the data support the conclusions?

Reviewer #1: Yes

3. Has the statistical analysis been performed appropriately and rigorously? 

Reviewer #1: Yes

4. Have the authors made all data underlying the findings in their manuscript fully available?

Reviewer #1: Yes

5. Is the manuscript presented in an intelligible fashion and written in standard English?

Reviewer #1: Yes

6. Review Comments to the Author

Reviewer #1: Since the comments made on earlier draft by me (and hopefully by other respected reviewers also) are attended positively/adequately, I am satisfied and, in my opinion, the manuscript is improved a lot. I recommend acceptance, without any hesitation, as now it has achieved acceptable level of our journal, in my opinion.

7. PLOS authors have the option to publish the peer review history of their article (what does this mean?). If published, this will include your full peer review and any attached files.

Reviewer #1: **Yes: **Dr. Sanjeev Sarmukaddam

---

## [Editor Report · Acceptance letter]

14 Jul 2020

PONE-D-20-06832R1 

The Impact of a Randomized Controlled Trial of a Lifestyle Intervention on Postpartum Physical Activity among At-Risk Hispanic Women: Estudio PARTO 

Dear Dr. Chasan-Taber:

I'm pleased to inform you that your manuscript has been deemed suitable for publication in PLOS ONE. Congratulations! Your manuscript is now with our production department. 

Kind regards, 

on behalf of

Dr. Noël C. Barengo 

Academic Editor

PLOS ONE